# Development of an Organ-Directed Exosome-Based siRNA-Carrier Derived from Autologous Serum for Lung Metastases and Testing in the B16/BL6 Spontaneous Lung Metastasis Model

**DOI:** 10.3390/pharmaceutics14040815

**Published:** 2022-04-07

**Authors:** Mai Hazekawa, Takuya Nishinakagawa, Masato Hosokawa, Daisuke Ishibashi

**Affiliations:** Department of Immunological and Molecular Pharmacology, Faculty of Pharmaceutical Sciences, Fukuoka University, Fukuoka 814-0180, Japan; tnishi0703@fukuoka-u.ac.jp (T.N.); hosokawa@fukuoka-u.ac.jp (M.H.); dishi@fukuoka-u.ac.jp (D.I.)

**Keywords:** siRNA, autologous exosome, glypican-3, melanoma, lung metastasis

## Abstract

Exosomes are nano-sized extracellular vesicles that are known to carry various messages to distant cells. It was recently reported that cancer-derived exosomes are orientated to metastatic organs. However, there are no reports on drug carrier development using autologous serum-derived exosomes in vivo. The purpose of this study was to deliver therapeutic siRNAs for melanoma lung metastases using autologous serum-derived exosomes. Primary tumors were induced by subcutaneously injecting melanoma cells into the hindlimbs of female C57BL/6 mice. Primary tumors were surgically removed on day 14. On day 21 after tumor removal, lung metastases were evaluated. Exosomes were isolated from serum collected from mice on days 0, 3, 7, 10, and 14 after primary tumor inoculation. After isolating serum exosomes, siRNA-loaded exosomes were prepared. siRNA-loaded exosomes were intravenously injected into the B16/BL6 spontaneous lung metastasis model mice on days 0, 3, 7, and 10 after tumor removal. siRNA-loaded exosomes prepared with autologous serum-derived exosomes significantly decreased the number of metastatic lung colonies. Autologous serum-derived exosomes, which have high organ accumulation, could potentially be used as efficient carriers of therapeutic siRNAs for melanoma patients with lung metastases.

## 1. Introduction

Exosomes have recently emerged as crucial components of the cell-to-cell and cell-to-matrix communication pathways that are related to tumor invasion and metastasis [1,2]. Exosomes are small membrane vesicles of endocytic origin that are released from various cell types into the extracellular environment after the fusion of multivesicular bodies with the plasma membrane [3]. Our previous study suggested that cancer-cell-derived exosomes carry non-coding RNAs involved in cancer metastasis to other cancer cells [4,5]. Other groups have reported that an ‘exosomal protein signature’ could identify melanoma patients at risk of metastasis to nonspecific distant sites. Furthermore, it was found that circulating tumor-derived exosomes may be useful not only to predict metastatic propensity, but also to determine sites of future metastases [6].

Small interfering RNA (siRNAs) recognize and degrade their complementary target mRNAs in a sequence-specific manner at the post-transcriptional level [7,8]. Using RNA interference (RNAi) [7,8,9] to suppress target gene expression has widespread potential therapeutic applications. siRNAs have the benefit of targeting genes that are specific to tumor cells, leaving healthy, non-tumor tissue unaffected. However, developing safe and efficient carriers for siRNAs delivery remains a formidable barrier to animal experiments, let alone clinical trials. 

We have previously reported the utility of siRNA-PLGA hybrid micelles, which are a polymetric micellar system of a biodegradable polymer targeting for specific cells in systemic circulation, as a siRNA delivery system [10,11,12]. Among the remaining challenges for this technology is the problem of endosomal escape from endocytosis by polymetric micelles and the cytotoxicity of the cationic polymer used for surface charge regulation. To solve these problems, we focused on exosomes to develop a siRNA carrier using particles with a lipid bilayer structure using membrane fusion that can avoid endocytosis. Furthermore, we will harness the unique properties of cancer-derived exosomes that allow for delivery to metastatic organs. Regarding a siRNA delivery system, exosomes have the advantages of being automatically transplantable to metastatic sites, the simplicity of introducing siRNAs into lipid bilayer membrane particles, and high permeability as an original cell-to-cell communication tool. Thus, exosomes could be suitable carriers for siRNAs.

This study aimed to design a highly safe and transplantable exosome-based formulation as a siRNA carrier that is not derived from in vitro cancer cell culture supernatants and with extremely low immunogenicity. The purpose of this study was to confirm the efficacy of autologous serum-derived exosomes preparations containing carrier-derived exosomes and assess the distribution of exosomes and siRNA in the body as a therapeutic agent for lung metastases.

## 2. Materials and Methods

### 2.1. Materials

Mouse Gpc3-specific siRNA and negative control (N.C.) siRNA (Silencer Select siRNA) were purchased from Thermo Fisher Scientific, Inc. (Waltham, MA, USA). The siRNA sequences were as follows: Gpc3 siRNA sense: 5′-GAG UCA GUC UUA GAC AUC AdTdT-3′ and Gpc3 siRNA antisense: 5′-UGA UGU CUA AGA CUG ACU CdTdT-3′. Alexa488- and Alexa647-labeled siRNAs were also synthesized by Thermo Fisher Scientific, Inc. Dacarbazine was purchased from Wako Pure Chemical Industries (Osaka, Japan). Anti-PD-1 antibody was purchased from Abcam (Cambridge, UK). Fetal calf serum (FCS) and Dulbecco’s Modified Eagle Medium (DMEM) were purchased from Wako Pure Chemical Industries (Osaka, Japan). Cell Counting Kit-8 (CCK-8) reagent was purchased from Dojindo Laboratories (Tokyo, Japan).

### 2.2. Purification and Identification of Exosomes

Exosomes were isolated with the PureExo Exosome isolation Kit (P110, Cosmo Bio Co., Ltd., Tokyo, Japan), in accordance with the manufacturer’s instructions and as previously described [4,5]. Nanoparticle tracking analysis (NTA; Nanosight, Meerbuech, Germany) was used to detect the size distribution of exosomes. We assessed the protein concentration of isolated exosomes in phosphate-buffered saline (PBS) using the BCA assay. Western blots were conducted to identify exosomes by detecting the expression of the exosome markers CD9, CD63, and CD81.

### 2.3. Preparation of siRNA-Loaded Exosomes

siRNA-loaded exosomes were prepared by transfection with Lipofectamine 2000 in accordance with the manufacturer’s instructions. In brief, 160 pmol of siRNA was mixed with Lipofectamine, and then the mixture was incubated at room temperature. After that, the mixture, including siRNA, was added to 160 pmol of exosomes from serum diluted in PBS.

### 2.4. Cell Culture

Murine melanoma cell lines (B16/BL6) were gifted from the RIKEN BRC cell bank. B16/BL6 cells were cultured in DMEM (Wako Pure Chemical Industries, Osaka, Japan) supplemented with 10% FCS in a 95% air, 5% CO_2_ atmosphere at 37 °C. 

### 2.5. Measuring GPC3 Levels in B16/BL6 Cells Treated with siRNA-Loaded Exosomes Derived from Serum

To measure relative GPC3 expression by Western blot, B16/BL6 cells were seeded in a 24-well plate at a density of 5 × 10^5^ cells per well, 24 h prior to adding siRNA-loaded exosomes. siRNA or siRNA-loaded exosomes were then added at 20, 40, and 80 pmol (amount of siRNA). After 48 h, the cells were lysed with 1% (*w*/*v*) Triton X-100 solution in PBS and centrifuged to remove cell debris.

### 2.6. Cell Viability Assay

The effect of Gpc3 knockdown on the proliferation of B16/BL6 cells was evaluated by the CCK-8 assay. In brief, 1 × 10^4^ cells in 100 µL of DMEM containing 10% FCS were seeded in 96-well plates 24 h before treatment. Cells were then treated with 50 µL of siRNA or siRNA-loaded exosomes for 48 h; control cells were treated with 50 µL of PBS. Then, 15 µL of CCK-8 reagent was added to each well, and the cells were further incubated for 2 h at 37 °C. Absorbances were measured with a microplate reader at the test and reference wavelengths of 450 and 655 nm, respectively, to evaluate relative cell viability.

### 2.7. Mice

Five-week-old C57BL/6JJcl mice obtained from CLEA Japan, Inc. (Tokyo, Japan) were housed in polypropylene cages with sawdust bedding at 24 ± 1 °C, 50 ± 10% humidity, and a 12 h/12 h light/dark cycle; food and water were available ad libitum. Procedures for animal care and use were consistent with the internationally accepted Guidelines for Keeping Experimental Animals, issued by the Government of Japan. The researchers received ethical approval (Permission number: 1908054 approved on 21 August 2019 and 1910082 approved on 24 October 2019) and training from the Fukuoka University Ethics Committee. 

### 2.8. The B16/BL6 Spontaneous Lung Metastasis Mouse Model

The B16/BL6 spontaneous lung metastasis mouse model was used to examine the effects of siRNA-loaded exosomes derived from autologous serum on tumor metastasis. The model was established by a similar method to those described in previous reports [13,14,15]. In brief, 1 × 10^5^ B16/BL6 cells were injected subcutaneously into the right hindlimb. On day 14, the primary tumors were surgically excised by amputating the right hind limb under anesthesia. On day 21 after primary tumor removal, the mice were euthanized, and lungs were excised to evaluate lung metastases. 

### 2.9. Distribution of Exosomes and siRNA Delivered by Exosomes in the B16/BL6 Spontaneous Lung Metastasis Mouse Model

An in vivo fluorescence imaging system (Clairvivo OPT Plus, Shimadzu Co., Kyoto, Japan) was used to assess the distribution of exosomes derived from autologous serum and their cargo siRNA. At the time of measurement, B16/BL6 spontaneous lung metastasis mice were anesthetized with isoflurane, and then placed in the chamber of the in vivo fluorescence imaging system. Fluorescence images (ICG: ex 785 nm/em 845 nm, Alexa647: ex 658 nm/em 710 nm) were then alternately taken from five directions, 24 h after intravenous injection of ICG-loaded exosomes only or Alexa647-labeled siRNA-loaded exosomes on day 21 after primary tumor removal. The fluorescence and luminescence images were recorded with the Clairvivo OPT software, version 3.0. Unless otherwise stated, the exposure time for all fluorescence measurements was 1 s. The intensity of fluorescence derived from the ICG-labeled exosomes or Alexa647-labeled siRNA was evaluated using the region of interest (ROI) analysis of the images. The same color scale was used in all images to measure fluorescence intensities.

### 2.10. Measuring the Anti-Metastatic Effects of siRNA-Loaded Exosomes in the B16/BL6 Spontaneous Lung Metastasis Mouse Model

#### 2.10.1. Evaluation of Exosome Collection Time to Use for siRNA Carrier

To assess exosome accumulation and activity against melanoma lung metastases, exosomes were isolated from serum collected on days 0, 3, 7, 10, and 14 after the subcutaneous transplantation of melanoma cells. Serum samples of 10 mice were pooled to isolate exosomes. After transfecting siRNAs into the exosomes, siRNA-loaded exosomes were treated as a siRNA-based nanocarrier in B16/BL6 spontaneous lung metastasis model mice. The siRNA-loaded exosomes were then used to treat the 10 mice that the blood was collected from (autologous transplantation). siRNA-loaded exosomes were injected intravenously via the tail vein into lung metastasis model mice on days 0, 3, 7, and 10 after primary tumor removal.

#### 2.10.2. Effect of Survival Rate and Anti-Metastasis Effects of siRNA Loaded Exosome Compared with Dacarbazine or Anti PD-1 Antibody

siRNA-loaded exosomes prepared using exosome isolated from serum collected on days 14 after the subcutaneous transplantation of melanoma cells were treated in B16/BL6 spontaneous lung metastasis model mice. siRNA-loaded exosomes (6 μg/kg) were injected intravenously via the tail vein into lung metastasis model mice on days 0, 3, 7, and 10 after primary tumor removal. Dacarbazine (10 mg/kg) was injected intravenously via the tail vein into lung metastasis model mice on days 1, 2, 3, 4 and 5 after primary tumor removal. Anti PD-1 antibody (1 mg/kg) was injected intravenously via the tail vein into lung metastasis model mice on days 1, and 15 after primary tumor removal. Survival rate was evaluated until days 28 after primary tumor removal, and the number of colonies in lung was counted on day 21 after primary tumor removal.

### 2.11. Western Blot Analysis

SDS-PAGE and immunoblotting were performed as described previously [10]. To detect protein, cells were lysed in a lysis buffer, while, for exosomes, the isolated solution from serum was used directly. Primary antibodies used in the experiments included: anti-GAPDH (1:10,000; Acris Antibodies, Inc., San Diego, CA, USA), anti-ICAM-1 (1:1000; Abcam, Cambridge, UK), anti-TGF-β1 (1:1000; ATGen Co., Ltd., Sungnam, Korea), and rabbit polyclonal antibodies against GPC3 (1:400; Abcam), VCAM-1 (1:1000; Abcam), VE-cadherin (1:1000; Cell Signaling Technology, Danvers, MA, USA), VEGF-A (1:1000; BioLegend, San Diego, CA, USA). Equal protein amounts of each exosome sample were applied per lane on the basis of BCA assay results.

### 2.12. Statistical Analysis

Values are expressed as the mean ± SD (*n* = 3 − 5). Data were evaluated for statistical significance using the Bonferroni test for differences among groups. Overall significance was determined with one-way ANOVA (repeated measures). A *p*-value < 0.05 was considered statistically significant.

## 3. Results

### 3.1. Size Distribution and Detection of Exosome Markers in Exosomes Isolated from Serum

The hydrodynamic sizes of exosomes derived from the serum of normal mice were analyzed using an NTA system. As shown in Appendix A, the mean diameter of the exosomes was between 155 and 257 nm. The particles were identified as exosomes because the exosome markers CD9 and CD63 were detected in each sample by Western blot. However, CD81 was not detected in this study.

### 3.2. Measuring Cell Permeability by Flow Cytometry

The fluorescence intensities of Alexa488-labeled siRNA-loaded exosomes were measured using flow cytometry to determine the efficacy of the cellular uptake of exosomes derived from serum as an siRNA carrier. The peak in the siRNA-loaded exosomes was shifted to the right for all cell lines compared with the peak in the siRNA solution (Figure 1). The peak in the siRNA-loaded exosomes not only shifted to the right, but the number of accounts was also lower. It is considered that the activation of siRNA suppressed the growth of cancer cells and reduced the number of cells. 

### 3.3. GPC3 Expression in B16/BL6 Cells Treated with siRNA-Loaded Exosomes

siRNA sequences that exhibited potent dose-dependent knockdown effects when delivered via transfection reagents were selected for subsequent experiments using exosomes as the siRNA carrier without transfection reagents. GPC3 expression in B16/BL6 cells treated with siRNA-loaded exosomes was then evaluated by Western blot. As shown in Figure 2, siRNA-loaded exosomes significantly suppressed GPC3 expression in a concentration-dependent manner compared with control cells. On the other hand, siRNA or exosome alone had no effect on GPC3 knockdown. From these results and Figure 1, the GPC3 knockdown effect of siRNA-loaded exosomes was caused by the siRNA activity, and the exosome was considered to promote the intracellular uptake of siRNA as a drug carrier.

### 3.4. Cell Viability Assay

Cell viability was evaluated to determine the effects of GPC3 downregulation on cell proliferation. As shown in Figure 3, treatment with siRNA-loaded exosomes significantly suppressed cell proliferation in a dose-dependent manner compared with control cells. 

### 3.5. Protein Levels in Exosomes

Protein quantification by Western blot was performed to determine the levels of protein expression in serum-derived exosomes, which change depending on the timing of blood collection. It is difficult to normalize the protein levels expressed in exosomes using housekeeping protein such as GAPDH or β-actin because exosome are granules not cells. Therefore, the amount of protein applied in each lane was unified in this study. The proportion of cancer-derived exosomes was expected to increase in blood samples in a time-dependent manner. The results of a previous report suggest that the accumulation of cancer-derived exosomes in metastatic destinations might be due to expression of an adhesion factor on the membrane surface [6]. Furthermore, it has been reported that cancer-derived exosomes express highly inflammatory factors [16]. Therefore, we focused on ICAM-1, VCAM-1, and VE-cadherin as adhesion factors and TGF-β1 as an inflammatory factor. ICAM-1, VCAM-1, VE-cadherin, and TGF-β1 levels were increased in serum-derived exosomes depending on the blood sampling time, as shown in Figure 4. VEGF-A was also evaluated in this study as one of the neovascularization markers, which are typical of metastasis tumors. In particular, VEGF-A levels were increased at days 10 and 14; this was correlated with the results for ICAM-1 and VE-cadherin. These data supported our suggestion that the proportion of cancer-derived exosomes in the exosomes isolated from serum increases with cancer-bearing time.

### 3.6. Anti-Metastatic Effects of siRNA-Loaded Exosomes in the B16/BL6 Spontaneous Lung Metastasis Mouse Model

We then examined the antitumor effects of siRNA-loaded exosomes in the B16/BL6 spontaneous lung metastasis mouse model. The average number of colonies in the lung (metastases) was 28.2 ± 8.3 21 days after primary tumor removal in control animals. The intravenous injection of siRNA-loaded exosomes significantly reduced the number of colonies in the lung as shown in Figure 5. A greater inhibitory effect was obtained with siRNA-loaded serum-derived exosomes that were collected from mice that had tumors for longer time periods. There were no significant differences between groups treated with siRNA solution or with exosomes without siRNA compared with non-treatment groups in the B16/BL6 spontaneous lung metastasis mice (data not shown).

### 3.7. Localization of ICG-Labeled Exosomes in the B16/BL6 Spontaneous Lung Metastasis Mouse Model

We examined the distribution of serum-derived exosomes after a single intravenous injection at 21 d after primary tumor removal in the B16/BL6 spontaneous lung metastasis mouse model. Images were obtained 24 h after the injection of each set of ICG-labeled exosomes. Greater accumulation in the lung was observed with the siRNA-loaded exosomes derived from serum collected from mice with a longer duration of tumor inoculation as shown in Figure 6. In fact, the labeled exosomes tended to accumulate in all organs as well as being increased in the lungs. We found that the amount of organ accumulation caused by serum-derived exosomes increased with the amount of time the mice carried tumors (Appendix A).

### 3.8. Localization of Alexa 647-Labeled siRNA-Loaded Exosomes in the B16/BL6 Spontaneous Lung Metastasis Mouse Model

The distribution of Alexa647-labeled siRNA delivered by exosomes was determined 21 d after primary tumor removal in the B16/BL6 spontaneous lung metastasis mouse model. As shown in Figure 7, the fluorescein signal caused by siRNA-loaded exosomes became wider, and this was more apparent with autologous serum from mice that had tumors for a longer duration. In particular, lungs with more metastatic colonies showed stronger fluorescence accumulation. 

### 3.9. Survival Rate and Anti-Metastatic Effects of siRNA-Loaded Exosomes in the B16/BL6 Spontaneous Lung Metastasis Mouse Model Compared with Dacarbazine and Anti-PD-1 Antibody

As shown in Figure 8A, the survival rate of the B16/BL6 spontaneous lung metastasis mouse treated with siRNA-loaded exosome was increased compared with dacarbazine or anti PD-1 antibody treatment groups, for which these drugs are used clinically as the standard of care at present. No deaths were observed during the experimental period in siRNA-loaded exosome treatment group. At the day 21 after the removal of primary tumor, the number of colonies in lung metastasis was significantly decreased in the siRNA-loaded exosome treatment group compared with the control group, as shown in Figure 8B.

## 4. Discussion

Previous studies have reported that the differences between exosomes and extracellular vesicles are defined by size and surface protein markers [17,18,19,20,21]. Exosome (30–300 nm) is smaller than extracellular vesicles (200–1000 nm). Furthermore, CD9, 63, 81 is expressed on the surface of exosome specificity. Our resulting particles, obtained using a commercial isolation kit, were identified as exosomes from the results of Appendix A. As shown in Appendix A, the exosomes obtained in this study tended to have a slightly larger SD in terms of article size distribution. It was suggested that SD may have increased due to the mixture of exosomes produced by various cell types in serum compared to the artificial exosomes or exosomes isolated from cell culture supernatant. 

One merit of autologous transplantation, which is the therapeutic concept of this study, is that immunogenicity would be low. Additionally, using autologous serum means collecting exosomes from individuals who already have cancer in their bodies. It is very likely that exosomes with higher organ accumulation can be collected compared to the use of exosomes derived from normal cells in vitro because organ accumulation is a property of cancer-derived exosomes isolated from serum. 

The levels of adhesion factors and inflammatory factors expressed by serum-derived exosomes increased according to the amount of time that tumor cells were inoculated in the mice. It was assumed that the proportion of cancer-cell-derived exosomes in the isolated exosomes also increased over time. This could suggest that the properties of cancer-derived exosomes influenced the altered protein concentrations in the total exosome pool. In fact, other studies have reported that B16/BL6 cells highly express each marker protein [22]. It is known that the composition of the membrane surface of exosomes depends on the properties of the secretory cells. It has been reported that integrins are involved in the organ-specificity of cancer-derived exosomes by analyzing different human cancers [6]. However, we did not observe a change in integrin 6α levels in blood collected at different timepoints in this study. It is possible that these contradictions are due to the studies being performed in humans and mice, respectively.

Exosomes consist of lipid bilayer membranes and could, therefore, be carriers for the intracellular delivery of siRNAs because they have high cell permeability and enhance the stability of nucleic acids. Furthermore, applying exosomes as siRNA carriers has the advantage of avoiding endosome escape via the endocytosis pathway, which remains a problem for polymeric micellar systems that use membrane fusion as the main intracellular uptake pathway. Stabilized siRNA using artificially synthesized nucleic acid or conjugate is one technique to improve delivery efficacy. The cytotoxicity of molecules used for stabilized siRNA remained. It is considered to be a safe carrier in terms of its low antigenicity because exosomes are biological components, especially autologous exosomes, in this study. A previous study indicated that exosomes can serve as potent therapeutic carriers since they offer low immunogenicity and high stability compared with other approaches [23]. This study also supported our ideas. It would also be advantageous to choose a material that can be artificially synthesized, such as artificial exosomes, as shown in previous reports [24]. Our analyses of exosome localization (Appendix A) showed that the serum-derived exosomes had increased organ accumulation and this feature was increased with an increased tumor inoculation time in mice; however, this organ accumulation was not lung-specific. In addition, when we compared the absolute values of accumulation, exosomes accumulated in organs with high blood flow. Thus, the further selection of exosomes and identification of surface proteins for specific accumulation is needed to develop a carrier with specific accumulation in the lung as a metastatic destination. However, a remarkable enhanced permeation and retention effect was also observed for exosomes in this study. In particular, the data on survival rate and anti-metastatic effects, which were shown to be superior to standard therapeutic agents, also determined the efficacy of this formulation as shown in Figure 8. There were some risks regarding cancer exosomes, which are thought to be involved in cancer metastasis. However, the prolongation of survival and suppression of the number of metastasis colonies in the lung proved that the strategy of this study was more effective than current treatments, with the benefits exceeding the risks. Thus, exosomes were shown to be useful as drug carriers to deliver siRNA into cancer cells. In this study, the application of serum-derived exosomes improved the efficiency of lung accumulation.

This study employed a unique strategy, in that it is the first in vivo experiment looking at a drug carrier using the autologous transplantation of self-serum exosomes in a B16/BL6 spontaneous lung metastasis mouse model. Our results suggest that blood containing a high number of cancer-derived exosomes can be collected immediately before excision surgery or after the discovery of the primary lesion to develop targeting agents. It has also been hypothesized that excision of the primary lesion, which is unavoidable in clinical practice, triggers the activation of metastasis. Furthermore, it has been reported that some non-coding RNAs in cancer-cell-derived exosomes can accelerate metastasis. Considering these reports, it is necessary to optimize the membrane components of exosomes with higher integration and find a pharmaceutical design that does not contain factors that promote cancer metastasis. Exosomes could be a more efficient targeting technique if future studies further identify the membrane surface proteins that define the metastatic destination.

Glypica-3 (GPC3) is a member of the heparan sulfate proteoglycans (HSPGs), attached to the cell surface by a glycosylphosphatidylinositol anchor. It has been reported to promote cell proliferation and metastasis and active inflammatory responses in melanoma [25], ovarian clear call carcinoma (OCCC) [26], lung cancer [27] and hepatocellular carcinoma (HCC) [28]. Therefore, it has been speculated that GPC3 knockdown could suppress cancer cell growth and metastasis. In our previous study, we discovered the utility of GPC3 knockdown in ovarian cancer [10]. Therefore, although the main purpose of this study was to develop a novel siRNA carrier, we also proved that GPC3 could be a target for therapies against metastatic melanoma. The therapeutic effects of GPC3 knockdown on melanoma metastasis could include the additive or synergistic effects of controlling not only cancer cells but also immune cells, because GPC3 is particularly involved in cancer cell growth, migration, infiltration, and in immune responses. On the basis of these points, this study revealed the utility of exosomes as siRNA carriers and the usefulness of anti-cancer treatments that involve knocking down GPC3 in vivo.

## 5. Conclusions

In this study, we verified the utility of exosomes as siRNA carriers in vivo. This study was the first to show the enhanced accumulation of exosomes at metastatic destinations by using autologous serum-derived exosomes, which are rich in cancer-derived exosomes. Thus, the use of autologous transplantation with formulations prepared from serum-derived components could be a novel and useful therapeutic strategy that offers low immunogenicity.

## Figures and Tables

**Figure 1 pharmaceutics-14-00815-f001:**
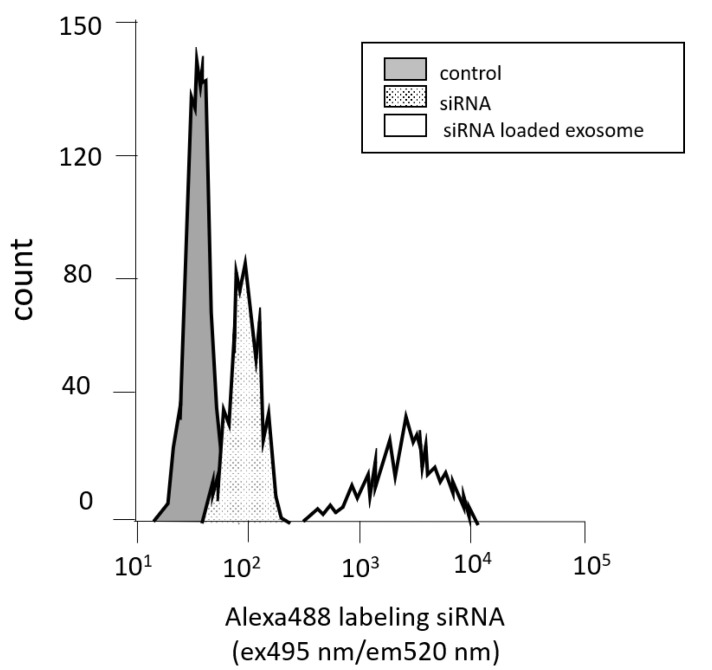
Assessment of cellular uptake by B16/BL6 cells treated with Alexa488-labeled siRNA-loaded exosomes derived from the serum of healthy mice using flow cytometry.

**Figure 2 pharmaceutics-14-00815-f002:**
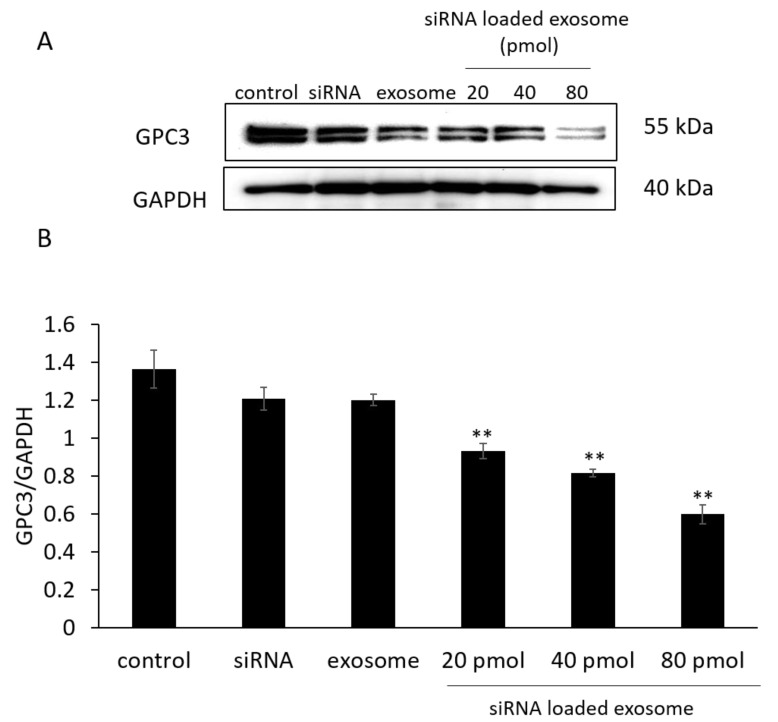
(**A**) Blotting images of GPC3 levels in B16/BL6 cells treated with siRNA-loaded exosomes at various siRNA concentrations and (**B**) its western blotting analysis. Data represent the mean ± SD of three independent experiments; ** *p* < 0.01 versus the control group (Bonferroni test/ANOVA).

**Figure 3 pharmaceutics-14-00815-f003:**
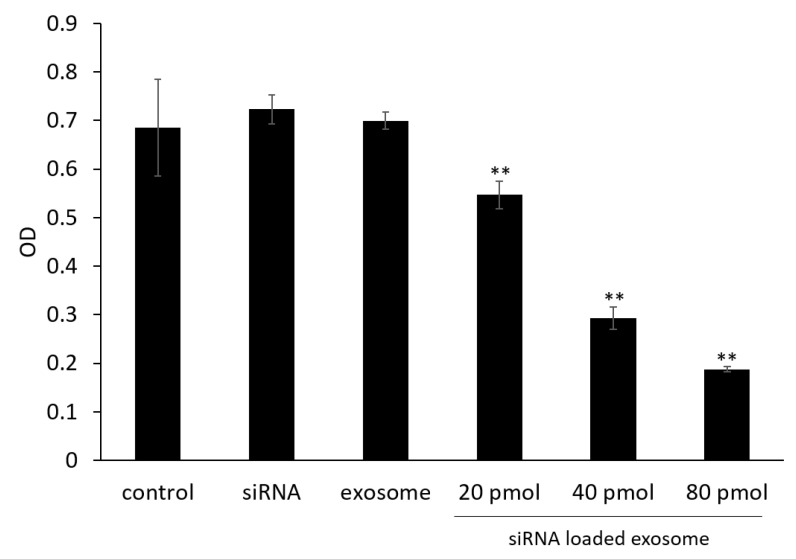
Proliferation rates of B16/BL6 cells treated with siRNA-loaded exosomes at various siRNA concentrations. Data represent the mean ± SD of three independent experiments; ** *p* < 0.01 versus the control group (Bonferroni test/ANOVA).

**Figure 4 pharmaceutics-14-00815-f004:**
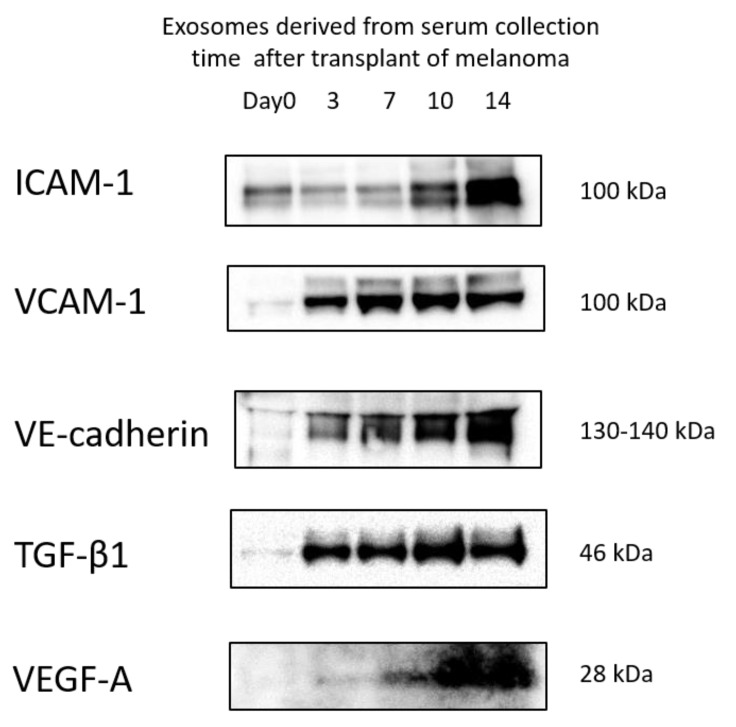
Changes in the levels of ICAM-1, VCAM-1, VE-cadherin, TGF-β1, and VEGF-A in exosomes isolated from serum at the indicated time points by western blotting analysis.

**Figure 5 pharmaceutics-14-00815-f005:**
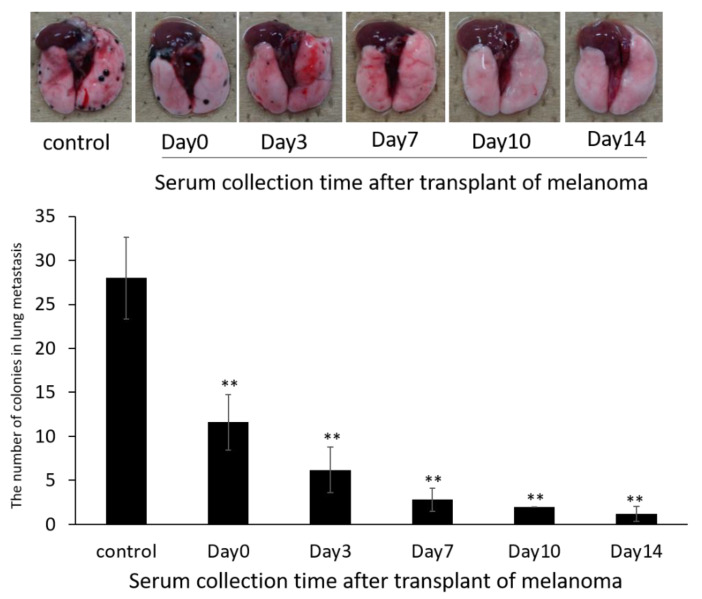
Anti-metastatic effects of siRNA-loaded exosomes prepared using exosomes derived from autologous serum collected at different timepoints from B16/BL6 spontaneous lung metastasis model mice. The number of colonies (lung metastases) were manually counted from images. Data represent the mean ± SD of five mice; ** *p* < 0.01 versus control B16/BL6 spontaneous lung metastasis model mice (Bonferroni test/ANOVA).

**Figure 6 pharmaceutics-14-00815-f006:**
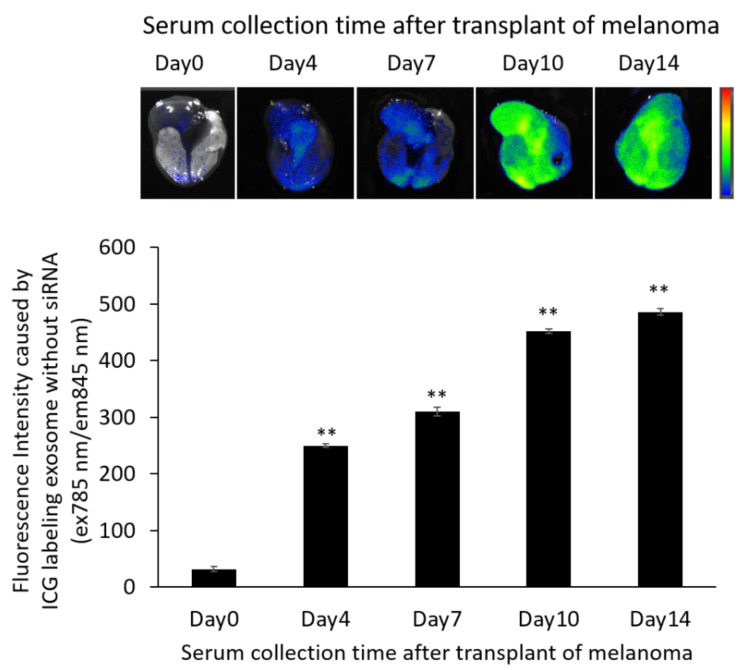
In vivo fluorescence imaging of the lung in B16/BL6 spontaneous lung metastasis model mice treated with ICG-labeled exosomes. Data were obtained 24 h after treatment with ICG-labeled exosomes. Data represent the mean ± SD of three independent experiments; ** *p* < 0.01 versus control B16/BL6 spontaneous lung metastasis model mice (Bonferroni test/ANOVA).

**Figure 7 pharmaceutics-14-00815-f007:**
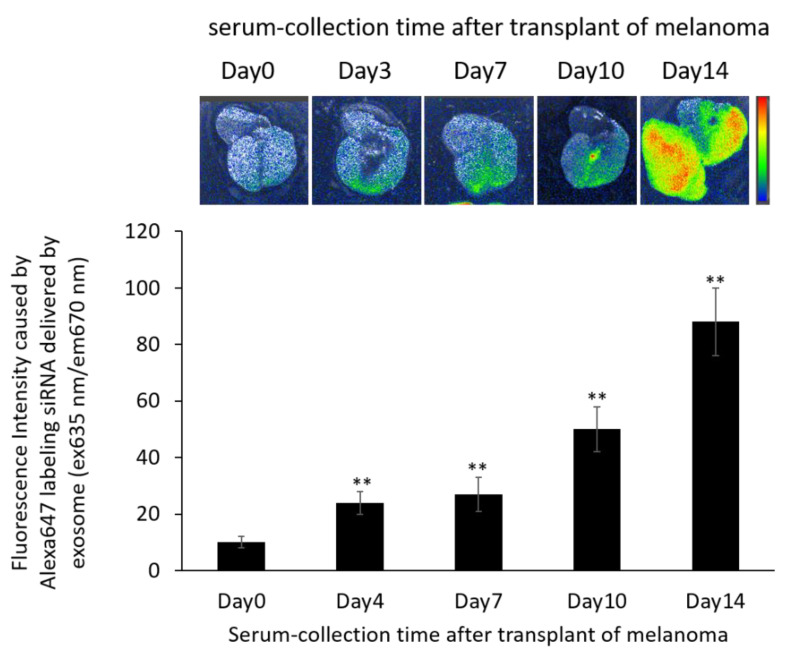
In vivo fluorescence imaging of the lung in B16/BL6 spontaneous lung metastasis model mice treated with Alexa647-labeled siRNA-loaded exosomes prepared from serum-derived exosomes. Data were obtained 24 h after treatment with Alexa647-labeled siRNA-loaded exosomes. Data represent the mean ± SD of three independent experiments; ** *p* < 0.01 versus control B16/BL6 spontaneous lung metastasis model mice (Bonferroni test/ANOVA).

**Figure 8 pharmaceutics-14-00815-f008:**
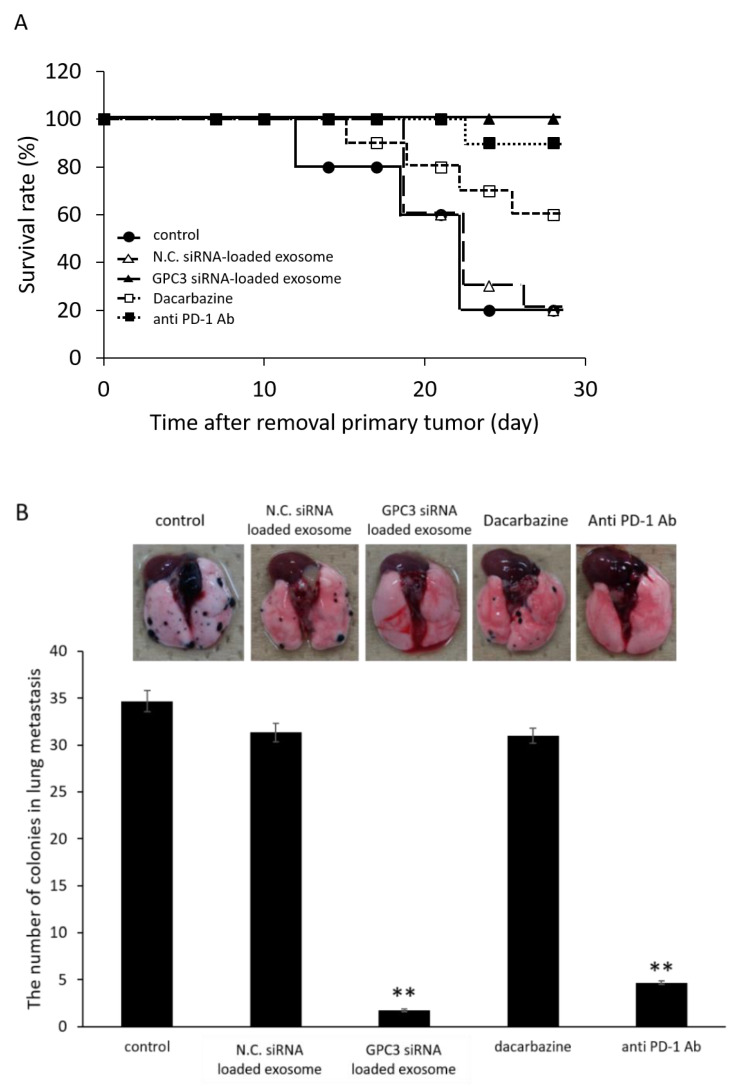
Survival rate (**A**) and anti-metastatic effects (**B**) in the lung metastasis of B16/BL6 spontaneous lung metastasis model mice treated with siRNA-loaded autologous exosome prepared with isolated from serum collected at days 14 after transplantation of B16/BL6 cells. Data represent the mean ± SD of five independent experiments; ** *p* < 0.01 versus control B16/BL6 spontaneous lung metastasis model mice (Bonferroni test/ANOVA).

## Data Availability

Not applicable.

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
