# Peer review of "Development of an Organ-Directed Exosome-Based siRNA-Carrier Derived from Autologous Serum for Lung Metastases and Testing in the B16/BL6 Spontaneous Lung Metastasis Model"

_pharmaceutics, 2022, doi:10.3390/pharmaceutics14040815_

Round 1

Reviewer 1 Report

The manuscript is a well written and organized article about the efficacy of autologous serum-derived exosomes and siRNA as a therapeutic agent for lung metastases. The work is well developed and have interesting results. Nevertheless, there are some points should be added to improve the manuscript.

  1. The figure S1 does not show the number of replications (n).

Is it very important that the standard deviation is so large in the size of exosomes on different days? What is this fact due to? Please, the authors should discuss this in the discussion.

  1. In section 3.2 The peak of the siRNA-loaded exosomes not only was shifted to the right but also the number of accounts was lower. Authors should explain that fact.
  2. In figure 2, authors should include the comparation between siRNA-loaded exosomes and both siRNA and exosome.
  3. In section 3.5 it is highly recommended that authors include neovascularization markers such as VEGF, which are typical of metastatic tumors.
  4. In section 3.6 the colony counting is performed from the macro images. It would be very convenient to paraffinize and cut the treated lungs to count the colonies in different slices along the lung.
  5. In section 3.7 and 3.8, is the fluorescence averaging done only in one section of the lung? What is the thickness of the fluorescence image?
  6. The authors should improve the discussion based on the suggestions made in the results.

Author Response

Reply to Reviewer #1

The manuscript is a well written and organized article about the efficacy of autologous serum-derived exosomes and siRNA as a therapeutic agent for lung metastases. The work is well developed and have interesting results. Nevertheless, there are some points should be added to improve the manuscript.

 Thank you for your thoughtful comments in order to make our manuscript better. The revised paragraph in the text was highlighted as written in red.

1. The figure S1 does not show the number of replications (n).

Is it very important that the standard deviation is so large in the size of exosomes on different days? What is this fact due to? Please, the authors should discuss this in the discussion.

→This result is the average and SD calculated for the values of exosomes of the same lot measured three times by NTA. It was suggested that SD may have increased due to a mixture of exosomes produced by various types of cells in serum compared to artificial exosomes or exosomes isolated from cell culture supernatant.

According to reviewer’s indication, our suggestion about SD was newly added in the discussion as follows.

Line 8 in page 17 in the revised text

As shown in Figure S1C, the exosomes obtained in this study tended to have a slightly larger SD in article size distribution. It was suggested that SD may have increased due to a mixture of exosomes produced by various types of cells in serum compared to artificial exosomes or exosomes isolated from cell culture supernatant.

2. In section 3.2 The peak of the siRNA-loaded exosomes not only was shifted to the right but also the number of accounts was lower. Authors should explain that fact.

→It is considered that the activation of siRNA suppressed the growth of cancer cells and reduced the number of cells.

According to reviewer’s suggestion, our consideration about the fact was newly added in the results as follows.

Line 12 in page 12 in the revised text

The peak of the siRNA-loaded exosomes not only was shifted to the right but also the number of accounts was lower. It is considered that the activation of siRNA suppressed the growth of cancer cells and reduced the number of cells.

3. In figure 2, authors should include the comparation between siRNA-loaded exosomes and both siRNA and exosome.

→According to reviewer’s comments, explanation about results was newly added in the results as follows.

Line 6 in page 13 in the revised text

On the other hand, siRNA or exosome alone had no effect on GPC3 knockdown. From these results and Figure 1, the GPC3 knockdown effect of siRNA-loaded exosomes was caused by the activity of siRNA, which was considered that exosome promoted intracellular uptake of siRNA as a drug carrier.

4. In section 3.5 it is highly recommended that authors include neovascularization markers such as VEGF, which are typical of metastatic tumors.

→As reviewer recommended, VEGF-A was newly evaluated in this study as one of neovascularization marker. These additional data supported our suggestion that the proportion of cancer-derived exosomes in the exosomes isolated from serum increases with cancer-bearing time.

According to reviewer’s comments, the data of VEGF-A levels in exosomes was newly added in the results as follows.

Line 10 in page 14 in revised text

VEGF-A was also evaluated in this study as one of neovascularization marker, which are typical of metastasis tumors. In particular, VEGF-A levels were increased at days 10 and 14, it was correlated with the result of ICAM-1 and VE-cadherin. These data supported our suggestion that the proportion of cancer-derived exosomes in the exosomes isolated from serum increases with cancer-bearing time.

5. In section 3.6 the colony counting is performed from the macro images. It would be very convenient to paraffinize and cut the treated lungs to count the colonies in different slices along the lung.

→Thank you for your accurate advice on how to count the colonies to improve our evaluation. In future studies, we would like to consider an evaluation method using different slices of paraffinized sample.

6. In section 3.7 and 3.8, is the fluorescence averaging done only in one section of the lung? What is the thickness of the fluorescence image?

→Fluorescence intensity was measured by ROI analysis using macro images of the entire lung without slicing as shown in the upper part of Figure 6 and Figure

7. The authors should improve the discussion based on the suggestions made in the results.

→Thank you for your thoughtful comments in order to improve our manuscript. According to the advice of the three reviewers including you, we have newly added describe as a discussion based on our results.

The END

Reviewer 2 Report

Dear authors

In the manuscript, autologous exosomes were utilized to deliver GPC3 siRNA against melanoma lung metastasis. The idea is that exosomes produced after tumor implantation could deliver efficiently siRNA to the tumors.

Cancer exosomes could target metastatic sites or prepare the metastatic niche for a next way of metastasis. Intrinsically, cancer exosomes could have dangerous effects. Following my considerations.

Major points

Have you compared exosomes isolated from normal serum vs “cancer” serum? Is there any significant difference?

Fig 4. There is no normalization

Fig 5 What is the advantages of exosomes versus others approaches? For examples, if authors use stabilized siRNA are inferior or superior to exosome-siRNA strategy?

Minor points

Please evaluate the guideline for exosome isolation to decide if to call for exosomes or extracellular vesicles.

Please check the references, some data are missing

Fig S2 Kidn(e)y

Author Response

Reply to Reviewer #2

In the manuscript, autologous exosomes were utilized to deliver GPC3 siRNA against melanoma lung metastasis. The idea is that exosomes produced after tumor implantation could deliver efficiently siRNA to the tumors.

Cancer exosomes could target metastatic sites or prepare the metastatic niche for a next way of metastasis. Intrinsically, cancer exosomes could have dangerous effects. Following my considerations.

 →Thank you for your thoughtful comments and suggestions. As you pointed out the dangerous effects of cancer exosome, it is important issues to apply exosomes to drug carrier for the treatment cancer. However, the novelty of this study is that a sufficient therapeutic effect of siRNA-loaded autologous exosomes was obtained in mice model as one targeting drug delivery technique. We understand that the next challenges are the design of a drug formulation using exosome that eliminates metastasis-promoting factor as we have already described in line 10 in page 20 in the revised text as follows.

Line 10 in page 20 in the revised text

Considering these reports, it is necessary to optimize the membrane components of exosomes with higher integration and find a pharmaceutical design that does not contain factors that promote cancer metastasis.

According to reviewer’s suggestion, our manuscript was revised. The revised paragraph in the text was highlighted as written in red.

Major points

Have you compared exosomes isolated from normal serum vs “cancer” serum? Is there any significant difference?

→We have compared exosomes from normal serum with cancer serum to evaluate isolation efficacy, transfection efficacy of siRNA and cellular uptake efficacy. There were no significantly differences in transfection efficacy of siRNA and cellular uptake efficacy. While, the amount of exosome isolated from cancer serum was larger than exosome isolated from normal serum significantly. Regarding our data of pharmacokinetics of exosomes in this study, it was considered that exosome isolated from serum at day 0 in spontaneous lung metastasis mice model appear to be similar to normal exosomes.

Fig 4. There is no normalization

→It is difficult to normalize protein levels expressed in exosomes using housekeeping protein such as GAPDH or β-actin because exosome are granules not cells. Therefore, the amount of applied protein in each lane was unified in this study.

According to reviewer’s pointing out, the reason for not normalization was newly added in revised text as follows.

Line 18 in page 13 in the revised text

It is difficult to normalize protein levels expressed in exosomes using housekeeping protein such as GAPDH or β-actin because exosome are granules not cells. Therefore, the amount of applied protein in each lane was unified in this study.

Fig 5 What is the advantages of exosomes versus others approaches? For examples, if authors use stabilized siRNA are inferior or superior to exosome-siRNA strategy?

 →The advantage of exosome is that the problem of endosome escape efficiency after intracellular uptake by endocytosis can be eliminated as compared with polymetric micelles because of exosome composed of lipid bilayer. Exosome is achieved efficient intracellular uptake by non-endocytosis pathway such as membrane fusion as described in discussion (Line 12 in page 18). As reviewer suggested, stabilized siRNA using artificially synthesized nucleic acid or conjugate are one of the techniques to improve delivery efficacy. While, there was remained the cytotoxicity of molecules used for stabilized siRNA. It is considered to be a safe carrier in terms of low antigenicity because exosomes are biological components, especially autologous exosomes in this study. Previous study has indicated that exosomes can serve as potent therapeutic carriers since they offer low immunogenicity, high stability compared with others approaches [23]. This study also supported our ideas.

 According to reviewer’s suggestion, discussion about the advantage of exosome compared with others approach was added in terms of immunogenicity using references in discussion in revised text as follows.

Line 14 in page 18 in the revised text

Stabilized siRNA using artificially synthesized nucleic acid or conjugate are one of the techniques to improve delivery efficacy. While, there was remained the cytotoxicity of molecules used for stabilized siRNA. It is considered to be a safe carrier in terms of low antigenicity because exosomes are biological components, especially autologous exosomes in this study. Previous study has indicated that exosomes can serve as potent therapeutic carriers since they offer low immunogenicity, high stability compared with others approaches [23]. This study also supported our ideas.

Minor points

Please evaluate the guideline for exosome isolation to decide if to call for exosomes or extracellular vesicles.

→In previous studies, the difference between exosomes and extracellular vesicles was defined by size and surface protein markers. That is why, we evaluated tetraspanins such as CD9, 63,81 expression levels using western blotting and size distributions to identify the obtaining particles as exosome in Figure S1.

According to reviewer’s suggestions, explanations about identification the obtaining particles as exosome not to extracellular vesicles were newly added in discussion in the revised manuscript as follows.

Line 4 in page 17 in the revised text

Previous studies have reported that the differences between exosomes and extracellular vesicles is defined by size and surface protein markers [17-21]. Exosome (30-300 nm) is smaller than extracellular vesicles (200-1000 nm). Furthermore, CD9, 63, 81 is expressed on surface of exosome specificity. Our resulting particles using commercial isolation kit were identified as exosome from the results of Figure S1.

Please check the references, some data are missing

→I’m sorry, it was our missing. The references were collected in the revised text.

Fig S2 Kidn(e)y

→I’m sorry, it was also our missing. The spelling was collected as follows.

Figure S2

Kidny→Kidney

The END

Reviewer 3 Report

I overall really enjoyed reading the manuscript by Hazekawa et al, on there ongoing work with siRNA delivery in exosomes for treatment of malignancy. The authors made a compelling case for the use of GPC3 siRNA for treatment of metastatic melanoma in mouse models. While the authors demonstrated reduction in lung mets in the mice, I think an experiment examining mouse survival would be helpful to support the authors hypothesis. This would be more conclusive that the treatment was more effective.  

Author Response

Reply to Reviewer #3

I overall really enjoyed reading the manuscript by Hazekawa et al, on there ongoing work with siRNA delivery in exosomes for treatment of malignancy. The authors made a compelling case for the use of GPC3 siRNA for treatment of metastatic melanoma in mouse models. While the authors demonstrated reduction in lung mets in the mice, I think an experiment examining mouse survival would be helpful to support the authors hypothesis. This would be more conclusive that the treatment was more effective.  

Thank you for your thoughtful comments in order to make our manuscript better. The revised paragraph in the text was highlighted as written in red.

According to reviewer’s comments, survival rate and anti-metastasis data compared with standard therapeutic regents were newly added in our manuscript as Figure 8A and 8B. As addition of Fugure8, methods, results and discussion were also added in the revised text as follows.

Methods: Line 4 in page 10 in the revised text

2.10.2 Effect of survival rate and anti-metastasis effects of siRNA loaded exosome compared with dacarbazine or anti PD-1 antibody.

siRNA-loaded exosomes prepared using exosome isolated from serum collected on days 14 after the subcutaneous transplantation of melanoma cells were treated in B16/BL6 spontaneous lung metastasis model mice. siRNA-loaded exosomes (6 μg/kg) were injected intravenously via the tail vein into lung metastasis model mice on days 0, 3, 7, and 10 after primary tumor removal. Dacarbazine (10 mg/kg) was injected intravenously via the tail vein into lung metastasis model mice on days 1, 2, 3, 4 and 5 after primary tumor removal. Anti PD-1 antibody (1 mg/kg) was injected intravenously via the tail vein into lung metastasis model mice on days 1, and 15 after primary tumor removal. Survival rate was evaluated until days 28 after primary tumor removal, and the number of colonies in lung was counted on days 21 after primary tumor removal.

Results: Line 10 in page 16 in the revised text

3.9 Survival rate and anti-metastatic effects of siRNA-loaded exosomes in the B16/BL6 spontaneous lung metastasis mouse model compared with dacarbazine and anti PD-1 antibody.

As shown in Figure 8A, the survival rate of the B16/BL6 spontaneous lung metastasis mouse treated with siRNA-loaded exosome was increased compared with dacarbazine or anti PD-1 antibody treatment groups, which these drugs are used clinically as standard of care currently. No deaths were observed during the experimental period in siRNA-loaded exosome treatment group. At the days 21 after removal primary tumor, the number of colonies in lung metastasis was significantly decreased in siRNA-loaded exosome treatment group compared with control group as shown in Figure 8B.

Discussion: Line 12 in page 19 in the revised text

In particular, the data of survival rate and anti-metastatic effects superior to standard therapeutic agents also determined the efficacy of this formulation as shown in Figure 8. There were some risks of cancer exosomes, which are thought to be involved in cancer metastasis. However, the prolongation of survival and the suppression of the number of metastasis colonies in the lung proved that the strategy of this study was effective than current treatments, exceeded the risk.

The END

Round 2

Reviewer 1 Report

I accept with the changes.